# Imitation Learning Using Generalized Sliced Wasserstein Distances

## Abstract

Imitation learning methods allow to train reinforcement learning policies by way of minimizing a divergence measure between the state occupancies of the expert agent and the novice policy. Alternatively, a true metric in the space of probability measures can be used by invoking the optimal transport formalism. In this work, we propose a novel imitation learning method based on the generalized form of the sliced Wasserstein distance, which presents a number of computational and sample complexity benefits compared to existing imitation learning approaches. We derive a per-state reward function based on the approximate differential of the $\mathcal{SW}2$ distance which allows the use of standard forward RL methods for policy optimization. We demonstrate that the proposed method exhibits state-of-the-art performance compared to established imitation learning frameworks on a number of benchmark tasks from the MuJoCo robotic locomotion suite.

## 1 Introduction

Imitation learning methods aim to recover a policy based on a dataset of trajectories sampled from a policy or a human expert assumed to perform the task at a proficient level. Most existing methods can be described from a distribution matching perspective as a minimization of a distance or divergence measure objective defined between the state occupancy measure of the expert trajectories and the state occupancy measure of the policy induced via interaction with the environment. More specifically, these methods rely on estimating instances of the class of $f$-divergences Csiszár (1972); Ho & Ermon (2016); Fu et al. (2017); Ni et al. (2021) or integral probability metrics Sriperumbudur et al. (2009) such as the Wasserstein distance. The Wasserstein distance is an appealing objective for distribution matching due to its metric properties but the computation required for its estimation scales poorly in the number of samples and dimensions. Furthermore, the application of Wasserstein distance to the domain of imitation learning poses challenging due to the necessity of formulating a per-step reward function in order to perform policy optimization, which requires the full transport plan to be computed beforehand. While it has been shown that this can be circumvented by using a greedy approximation of the transport plan Dadashi et al. (2020) or utilizing a the Kantorovich-Rubinstein dual Arjovsky et al. (2017); Xiao et al. (2019), optimization remains a challenge.

Sliced optimal transport methods approximate the Wasserstein-2 distance by leveraging the increasing arrangements Santambrogio (2015) transport map which yields a closed form solution for the one-dimensional Wasserstein distance computation. The sliced Wasserstein ($\mathcal{SW}_2$) distance is computed by averaging over a number of randomly sampled one-dimensional projections (slices) of the empirical data distribution. It carries a number of computational benefits, in particular polynomial sample convergence properties Deshpande et al. (2019) as opposed to the Wasserstein distance, as well as worst-case log linear computational complexity. While the number of necessary slices has been shown to scale poorly with increasing dimensionality of the problem, a number of methods Deshpande et al. (2019); Nguyen et al. (2020); Kolouri et al. (2019a) have been developed to alleviate this issue. In particular, extensions such as the max-SW distance Deshpande et al. (2019) or the distributional SW distance Nguyen et al. (2020) have been shown to be effective.

In this work, we propose to leverage the generalized $\mathcal{SW}_2$ distance proposed in Kolouri et al. (2019a) for the purposes of imitation learning. For the purposes of sample efficiency, we adopt an off-policy formulation of the policy optimization procedure and discuss the implications in terms of the objective matching samples from policy mixtures arising in an off-policy optimization setting. We derive a

per-state reward function via a simple array insertion scheme using a sorting algorithm and show that the optimization of this reward expectation under the replay buffer distribution corresponds to the minimization of the sliced Wasserstein distance gradient.

Our main contribution is a novel imitation learning algorithm based on the minimization of the generalized sliced Wasserstein distance, which is a tractable optimal transport metric under mild assumptions on the functional properties of the nonlinear projection. To validate our method, we perform an empirical study of the performance of the algorithm on a number of benchmark tasks from the MuJoCo suite, where we observe improved sample efficiency and asymptotic performance compared to a number of adversarial imitation learning baselines.

## 2 BACKGROUND AND PROBLEM SETTING

This section introduces the necessary notation and formalism to describe the problem setting.

**MDP**  We consider environments modelled by a *Markov decision process* $M = (\mathcal{S}, \mathcal{A}, \mathcal{T}, \mu_0, R)$, where $\mathcal{S}$ is the state space, $\mathcal{A}$ is the action space, $\mathcal{T}$ is the family of transition distributions on $\mathcal{S}$ indexed by $\mathcal{S} \times \mathcal{A}$ with $p_\tau(s'|s,a) : \mathcal{S} \times \mathcal{A} \to \Delta(\mathcal{S})$ describing the probability of transitioning to state $s'$ when taking action $a$ in state $s$ using transition function $\tau(s,a) : \mathcal{S} \times \mathcal{A} \to \mathcal{S}$, $\mu_0$ is the initial state measure, and $R : \mathcal{S} \times \mathcal{A} \to \mathbb{R}$ is the reward function. A policy $\pi : \mathcal{S} \times \mathcal{A} \to \Delta(\mathcal{A})$ is a map from states $s \in \mathcal{S}$ to distributions $\pi(\cdot|s)$ over actions, with $\pi(a|s)$ being the probability of taking action $a$ in state $s$.

**Inverse reinforcement learning (IRL)**  IRL methods aim to estimate a suitable reward function $r_\psi$ parameterized by weights $\psi$ based on a dataset of expert trajectories $\mathcal{D}_E = \{\xi_i\}_{i \leq N}$ where $\xi_i = (s_{1:T}^{(i)}, a_{1:T}^{(i)})$ is a sequence of states and actions of expert $i$ of length $T$. To achieve this goal, the expectation statistics of the state features under the state occupancy measure of the expert, $\mathbb{E}_{\tau \sim \mathcal{D}_E}[\phi(\xi)]$, are matched with the statistics of the student state occupancy measure $\mathbb{E}_{\xi \sim p(\xi|\psi)}[\phi(\xi)]$. $\phi(\xi)$ denotes the feature representation of the trajectory $\xi$. The trajectory distribution $p(\xi|\psi)$ is induced by the policy $\pi_{r_\psi}$ trained on the reward function estimate $r_\psi$.

**Imitation learning via distribution matching**  Imitation learning (IL) methods are employed in MDP settings without access to an explicit reward function. IL methods are closely related to the IRL approach, but omit the explicit estimation of a stationary reward. Instead, they aim to directly minimize a divergence measure, typically an instance of the class of $f$-divergences or integral probability metrics between the empirical state occupancy measure $\rho_e = \sum_{i \leq N} \delta_{s_t^E}$ and policy occupancy measure $\rho_\pi = \sum_{i \leq N} \delta_{s_t^\pi}$. The optimization is performed w.r.t to the parameters of the policy, making use of the fact that there exists a one-to-one correspondence between the policy $\pi$ and the induced state occupancy measure $\rho_\pi$ Puterman (2014).

$$\mathcal{L}(\pi) = \underset{\pi}{\operatorname{argmax}} \, D(\rho_\pi, \rho_e)$$

**Optimal transport**  The optimal transport formalism allows the definition of a distance between two densities defined on the appropriate measure spaces $\rho \in \mathcal{M}(X)$ and $\varrho \in \mathcal{M}'(X)$. For any $p \geq 1$, the $p$-Wasserstein distance $W_p$ is defined as follows:

$$W_p(\rho, \varrho) = \left( \inf_{\nu \in \Gamma(\rho, \varrho)} \int_{\mathcal{X} \times \mathcal{X}'} c(x, x') d\nu(x, x') \right)^{\frac{1}{p}}$$

where $\Gamma(\rho, \varrho)$ is the set of all couplings between the densities $\rho$ and $\varrho$. For $d = 1$, the 2-Wasserstein distance $W_2$ admits a closed form obtained via sorting the samples of empirical distributions and calculating the total distance between them.

$$W_2(\rho, \varrho) = \int_0^1 |F_\rho^{-1}(\mu) d\mu - F_\varrho^{-1}(\mu)|^2 d\mu$$

where $F_\rho$ and $F_\rho^{-1}$ are the cumulative distribution function (CDF) and quantile (inverse CDF) function respectively. This corresponds to the *increasing arrangements* optimal transport map

$T(x) = (F_\rho^{-1}(\nu) \circ F_\varrho^{-1}(\mu))(x)$ Santambrogio (2015). The sliced Wasserstein distance $\mathcal{SW}_2(\rho, \varrho)$ is defined as the integral over 1-dimensional random slices defined via the Radon transform $\mathcal{R}\rho(t, \theta) = \int_{\mathcal{X}} \rho(x)\delta(t - \langle x, \theta \rangle)dx$ of the density $\rho(x)$:

$$\mathcal{SW}_2(\rho, \varrho) = \int_{\mathcal{S}^{d-1}} W_2(\theta_\# \rho, \theta_\# \varrho)d\theta \tag{1}$$

where $\theta_\# \rho$ denotes the pushforward measure of slice $\theta$. The challenge with estimating (1) is the computation of the integral over sphere directions $\theta \in \mathcal{S}^{d-1}$, requiring a Monte Carlo scheme. In particular, a high number of linear projections is often required for accurate approximation of the $\mathcal{W}^2$ distance due to the slow convergence rate $n^{-1/d}$ of the $\mathcal{W}^2$ estimator for a d-dimensional dataset of $n$ samples. The true $\mathcal{W}_2$ distance is bounded above and below: $SW_2(\rho, \varrho) \leq W_2(\rho, \varrho) \leq \alpha SW_2(\rho, \varrho)^\beta$, where $\alpha$ is a constant and $\beta = (2(d+1))^{-1}$ is a dimension dependent exponent Santambrogio (2015).

**Generalized Sliced Wasserstein Distances** In order to overcome the poor scaling properties of linear projections, Kolouri et al. propose to use nonlinear projections defined via the generalized Radon transform $\mathcal{G}\rho(t, \theta) = \int_{\mathcal{X}} \rho(x)\delta(t - g_\theta(x))dx$ where $g_\theta(x)$ is a nonlinear projection defined as a homogeneous polynomial of odd degree or parametrized by a neural network with parameters $\theta$. The corresponding generalized distance $\mathcal{GSW}_2$ is given as follows:

$$\mathcal{GSW}_2(\rho, \varrho) = \int_{\Omega_\theta} W_2(g_\theta(\rho), g_\theta(\varrho))d\theta \tag{2}$$

In order for the metric properties of the $\mathcal{GSW}_2$ distance to be satisfied, the injectivity of the generalized Radon transform is required, which is hard to satisfy in practice when using neural networks as defining functions for the Radon transform. This renders the $\mathcal{GSW}_2$ distance a pseudo-metric.

## 3 SLICED WASSERSTEIN IMITATION LEARNING (SWIL)

In this section, we describe our main algorithmic contribution: the sliced-Wasserstein imitation learning (SWIL) algorithm and discuss the specific implications of using sliced Wasserstein distances in the context of Markov decision processes.

### 3.1 SLICED WASSERSTEIN DISTANCES BETWEEN STATE OCCUPANCY MEASURES

As a first step, we define the state occupancy measures $\rho_e$ and $\rho_\pi$ we would like to compare. The expert occupancy measure corresponds to the empirical measure $\rho_e = \sum_{x \in \mathcal{D}_E} \delta_x$ of the dataset of expert demonstrations $\mathcal{D}_E$. The policy occupancy measure is defined analogously $\rho_\pi = \sum_{x \in \mathcal{D}_\pi} \delta_x$ where $\mathcal{D}_\pi$ is the rollout buffer of the policy interacting with the environment and $\delta_x = \mathbb{1}(X = x)$ is the characteristic function of set $X \subseteq \mathcal{X}$. [1]

The main goal of the algorithm is to learn a policy $\pi$ which induces a state occupancy measure $\rho_\pi \in \mathcal{M}(\mathcal{X})$ minimizing the $\mathcal{GSW}_2$ distance to the empirical state occupancy measure of the expert $\rho_e \in \mathcal{M}(\mathcal{X})$. We assume that both are defined over the same measure space $\mathcal{M}(\mathcal{X})$ and can thus be compared. In order to compute the $\mathcal{GSW}_2$ distance, we define a generalized projection network $g_\psi : \mathcal{X} \to \mathbb{R}^K$ mapping state-action tuples to $K$ projection directions. The network defines the pushforward measures $\psi_\# \rho_e$ and $\psi_\# \rho_\pi$ for which a Monte Carlo estimate of the $\mathcal{SW}_2$ distance based on the $K$ projection directions.

$$\mathcal{GSW}_2^\psi(\rho_e, \rho_\pi) = \frac{1}{K} \sum_{k \leq K} W_2(\psi_\# \rho_e, \psi_\# \rho_\pi) \tag{3}$$

relying on the closed-form one-dimensional $W_2$ computation via the increasing arrangements optimal transport map. While more expressive than linear projections, generalized projections also suffer

---

[1] Here, $\mathcal{X} \in \{\mathcal{S}, \mathcal{S} \times \mathcal{A}, \mathcal{S} \times \mathcal{S}, \mathcal{S} \times \mathcal{A} \times \mathcal{S}\}$ corresponds to the diffent variants of the underlying measurable space, defined as respective Cartesian products of state and action spaces.

from the issue that some projection directions are not informative. In order to circumvent this issue, we use the max-$\mathcal{GSW}_2$ distance parametrized by a neural network with parameters $\psi$ which replaces the sum in 3 with the maximum over the generalized projection directions and also constitutes a valid metric. The resulting optimization problem is then given as follows:

$$\mathcal{L}_{SWIL}(\pi, \psi) = \min_{\pi} \max_{\psi} \mathbb{E}_{\pi}[\mathcal{GSW}_2^{\psi}(\rho_e, \rho_{\pi})] \tag{4}$$

A direct minimization of the $\mathcal{GSW}_2$ distance via gradient descent requires derivatives w.r.t to the transition dynamics $\tau(s, a)$ which are only implicitly available. This formulation therefore necessitates the use of an adversarial optimization scheme. In order to obtain the projections necessary to compute the max-$\mathcal{GSW}_2$ distance, we employ a similar scheme to Deshpande et al. (2019) which defines a *linear discriminant projection subspace*, where the samples from $\rho_e$ and $\rho_{\pi}$ are easily distinguishable. In order to find this subspace, a discriminator $D(x) := (\psi \circ \varphi)(x)$ is trained using a binary classification loss in order to produce a feature representation $\varphi(x)$ and linear predictor $\psi$. Calculating the 1-D $W_2$ distance in this subspace has been shown to be upper bounded by the max-$\mathcal{SW}_2$ distance Deshpande et al. (2019). It has also been shown that the surrogate loss used in this scheme can be employed to approximately recover the maximum projection direction. Over the course of the training with $N$ update rounds, this results in a sequence of projections $(\psi_n)_{n \leq N}$ which allows the definition of $N$ 1-dimensional Wasserstein distances $W_2((\psi_n \circ \varphi)_{\#}\rho_e, (\psi_n \circ \varphi)_{\#}\rho_{\pi})$. We discuss the implications of this fact in 3.4. In order to perform the minimization w.r.t. the policy in 4, we propose to use a soft actor-critic $SAC$ agent Haarnoja et al. (2018).

**Using GSW in MDP** In order to apply the GSW distance in the context of imitation learning in Markov decision processes, it is necessary to provide a reward signal to the agent which is defined for every timestep. A natural candidate would be the GSW distance itself, however, defining it for a single state requires an unbalanced optimal transport Séjourné et al. (2023) formulation due to violation of mass preservation constraints. Instead, we propose to use a reward function which is based on a differential signal with respect to the GSW distance.

## 3.2 REWARD FORMULATION

In order to define the per-timestep reward required for policy optimization, we employ the following scheme. Let $\hat{\rho}_e$ and $\{\hat{\rho}_k\}_{k \leq K}$ be empirical samples drawn i.i.d. from the expert dataset and replay buffer respectively. The calculation of the $\mathcal{GSW}_2$ distance requires their projection on a set of 1-dimensional subspaces using the projection function $g_{\phi, \psi}(x) = (\psi \circ \phi)(x)$ and subsequent sorting of the projected atoms in the respective subspaces. As can be seen in Fig. 1, assuming equal number of atoms from expert and replay buffer, sorting the $n$ atoms allows the distance to be computed as the sum over the rectangle areas $\sum_{n \leq N} a_n$ where $a_n = \frac{1}{N} d(F_{\rho_k}^{-1}(1/n), F_{\rho_e}^{-1}(1/m))^2$ is the area of a single rectangle. Let $x_t \sim (\tau \circ \pi)(s_t)$ be a sample from the interaction of the policy $\pi$ with the environment using transition function $\tau$. We define the reward as change of integral when the newly sampled atom replaces the closest atom in $\hat{\rho}_k$. This requires the insertion (and replacement) of the new atom into the sorted set $\hat{\rho}_k$, which is an efficient computational routine ($\mathcal{O}(n)$). We define the reward as the *atom replacement loss*, where the sum is taken over multiple projections $K$:

$$r_{SW}(x_t) = -\frac{1}{K} \sum_{k \leq K} \Delta a_{\psi, n}(s_t) = -\sum_{k \leq K} \tilde{\nabla}_{x_t} W_2(g(\rho_k), g(\rho_e)) \quad x_t \in \mathcal{X} \tag{5}$$

## 3.3 ANALYSIS

In this section, we analyze the proposed reward computation scheme and show that optimizing its expectation over policy samples effectively minimizes the $\mathcal{SW}_2$ distance in the linear case. We assume $g$ to be identity and $K = 1$ for this part.

**Proposition 3.1.** *The agent's policy optimizing $\mathbb{E}_{\pi}[\sum_t \gamma^t r_{SW}(x_t)]$ minimizes the $\mathcal{SW}_2$ distance between the policy occupancy measure $\rho_{\pi}$ and expert occupancy measure $\rho_e$.*

---

[2]w.l.o.g. we assume $n = m$

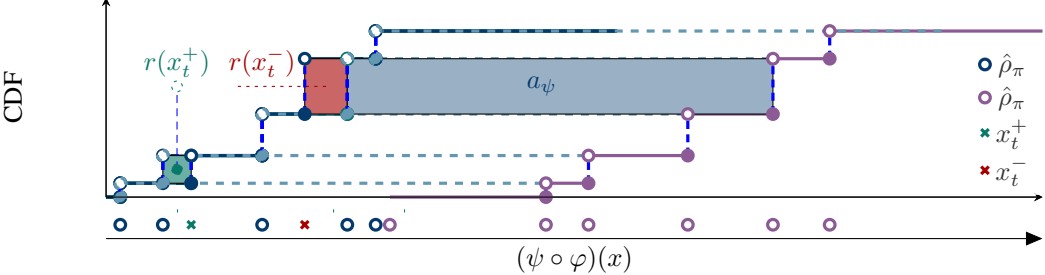

Figure 1: SWIL reward illustration. We propose a simple scheme where the reward for a state transition sampled from the environment is obtained by replacing the closest projected atom and computing the difference of rectangles $a_\psi$.

In figure 1, we can readily see that the reward calculation 5 corresponds to a finite difference approximation of the gradient of $\nabla_\pi W_2(\rho_\pi, \rho_e)$ for a single projection. The policy optimization effectively induces an approximation of a $\mathcal{SW}_2$ gradient flow Liutkus et al. (2019) on the space of occupancy measures, which is known to converge in the case of convex functions. In practice, for the purposes of sample efficiency, we make use of an off-policy RL algorithm to optimize the policy $\pi$.

**Off-policy distribution matching**   As opposed to on-policy adversarial imitation methods such as GAIL, the off-policy formulation poses an additional challenge in the case of distribution matching, since the samples are obtained by sampling the replay buffer, which contains samples from a mixture of previous policies. In its base form, this corresponds to minimizing a distance between the replay buffer occupancy measure $\rho_{RB}$ and expert occupancy measure $\rho_e$. This has been studied in Kostrikov et al. (2019); Zhu et al. (2020) in the context of learning from demonstrations and the learning from observations setting. It can be shown that by using the variational representation of the $f$-divergence, the optimization can be rewritten purely w.r.t to the critic function. In our case, we can make use of previous cached projections to regularize the policy behaviours.

## 3.4   DESIGN CHOICES

In this section, we outline a few of the design choices that need to be considered when deploying the algorithm in practice.

**Replacing atoms**   It is possible to get a tighter bound on the $\mathcal{GSW}_2$ distance by employing the "replacing atoms" strategy. This entails an online update of the $\mathcal{GSW}_2$ estimate at round $t$ by effectively replacing the atoms in the sample buffer $\hat\rho_k$ closest to the policy sample $x_t \sim (\tau \circ \pi)(s_t, a_t)$ if the distance decreases, i.e. if the reward is positive. This results in a "stricter" reward structure for subsequent policy samples and can promote faster convergence.

**Reusing cached samples**   The convergence of the procedure outlined in 3.2 can be slow and is prone to exhibit high variance. To combat the latter, we propose to reuse $K$ previous projections in order to compute the reward. Intuitively, this is similar to reusing the cached transitions in the replay buffer. We study the effect of these design choices empirically in 4.2.

**Motivation for choice of feature representation**   High-dimensional state spaces might have non-trivial topologies, which necessitate alternatives to the L2 cost typically used in the calculation of the Wasserstein distance by considering e.g optimal transport on manifolds Bianchini et al. (2011). This is a widely acknowledged fact in the image domain, however, this is also the case in various robotics scenarios, where the kinematic poses of objects typically feature a Cartesian product of Euclidean (translation) and non-Euclidean (rotation) geometries. The use of an intermediate feature representation $\varphi(x), x \in \mathcal{X}$ allows to defer this problem to the question of finding the optimal feature space, in which it is more amenable to use the L2 transport cost.

---

**Algorithm 1** Sliced Wasserstein imitation learning (SWIL)

---

**Input:** Expert trajectories $\{\xi_i\}_{i \leq N}$
Initialize soft actor-critic $\pi_\theta, Q_\vartheta$, projection network $g_{\psi,\varphi}$ and replay buffer $\mathcal{D}_{RB}$
**for** $t = 1$ **to** $N_{rounds}$ **do**
  Collect trajectory $\tau_i$ by executing the policy $\pi_\theta$ and add the transitions to $\mathcal{D}_{RB}$
  Update $g_{\phi,\psi}(x)$ using the surrogate binary cross entropy loss $\mathcal{L}_s$ between $\rho_e$ and $\rho_{RB}$ by
  minimizing

$$\mathcal{L}(\rho_e, \rho_{RB}) = \frac{1}{K} \sum_{k \leq K} \mathcal{L}_s(g_\psi(\rho_e), g_\psi(\rho_{RB}))$$

  using finite samples $(\hat{\rho}_e \sim \mathcal{D}_e, \hat{\rho}_\pi \sim \mathcal{D}_{RB})$
  Update the actor-critic functions $\pi_\theta, Q_\vartheta$ by using the SAC optimization procedure w.r.t to the
  *atom replacement loss* $r_\psi(s_t) = \frac{1}{|a|} \sum_{a_\psi} \Delta a_\psi(s_t)$ as the reward
**end for**

---

## 4 EXPERIMENTS

In order to evaluate our method empirically, we propose to conduct a set of experiments on a set of MuJoCo robot locomotion tasks. The experimental evaluation aims to answer the following questions:

- Can the proposed method recover the asymptotic performance demonstrated in the expert dataset in a sample efficient manner?

- How does the method perform in different *data sparsity* settings?

- What is the impact of different modeling choices on the performance of the algorithm?

**Baselines** We compare our method to a number of established baselines on MuJoCo locomotion tasks: Discriminator-Actor-Critic (DAC) Kostrikov et al. (2018), which is an off-policy GAIL-like algorithm which uses a Lipschitz regularizer, Primal Wasserstein Imitation Learning (PWIL) Dadashi et al. (2020), a primal OT problem solver which uses a greedy approximation of the transport plan as well as three variants of the GAIL algorithm: on-policy GAIL Ho & Ermon (2016) with spectral normalization Yoshida & Miyato (2017) [3], off-policy GAIL (with and without spectral normalization) and off-policy AIRL Fu et al. (2017), which uses a shaped discriminator structure. We use soft actor-critic (SAC) as the same off-policy forward RL algorithm for all baselines with the exception of on-policy GAIL which use the proximal policy optimization (PPO) Schulman et al. (2017) algorithm.

We further evaluate the methods in a number of *data sparsity regimes* by using different subsampling factors for the expert datasets. We conclude the experimental evaluation by investigating different aspects of the proposed algorithm by performing an ablation study on the design choices outlines in section 3.4.

In a first experiment, we compare the same algorithms in a more sophisticated set of benchmark environments. We select five robot locomotion tasks of moderate and high state space dimensionality ($d \in \{8, 11, 17, 111, 376\}$). [4] from the MuJoCo Todorov et al. (2012) suite. For each environment, we obtain the expert demonstrations by rolling out pretrained SAC models downloaded from the `stable-baselines3` repository on Huggingface [5]. We generate three sets of trajectories $|\mathcal{D}_E| \in \{1, 4, 10\}$ with 1, 4 and 10 trajectories in each dataset with a minimum ground truth cumulative reward threshold to ensure valid behaviours. Figure 2 depicts the training behaviour of the algorithm over 1M timesteps compared to the baselines for a different number of trajectories $|\mathcal{D}_E|$. We can observe that our method exhibits superior *sample efficiency* in four out of five tasks with the exception of `Hopper-v3` and demonstrates superior *asymptotic performance* on all tasks when trained on a single trajectory. The final rollout performance at 1M timesteps is further summarized in Table 1. As the number of trajectories increases, our algorithm exhibits a deteriorated asymptotic performance on the `HalfCheetah-v3` task. We hypothesize that this might be due to the more diverse set of

---

[3]here, we omit the default on-policy GAIL formulation due to poor performance
[4]`Ant-v3`, `HalfCheetah-v3`, `Hopper-v3`, `Walker2d-v3` and `Humanoid-v3`
[5]https://huggingface.co/sb3

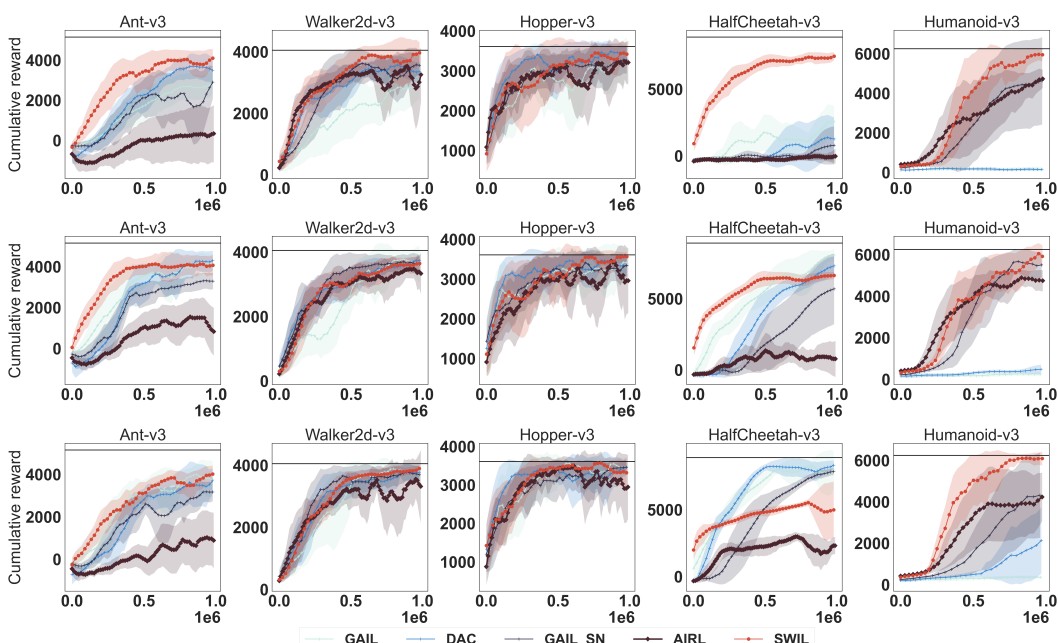

Figure 2: Comparison of imitation learning algorithms on five MuJoCo benchmark tasks. The three rows correspond to models trained on 1, 4 and 10 trajectories respectively.

demonstrations, which might necessitate a multi-marginal formulation of the sliced optimal transport problem Cohen et al. (2021).

## 4.1 DATA SPARSITY PERFORMANCE

In a second experiment, we investigate the behaviour of the proposed approach in a number of *data sparsity regimes*, induced by varying the subsampling factor of a single expert demonstration $\varsigma \in \{1, 20, 100\}$. We perform a comparison against baselines on two of the more challenging robot locomotion tasks: Ant-v3 and Humanoid-v3. The performance is illustrated in Figure 4. Remarkably, our algorithm is able to recover the expert behaviour on the challenging Humanoid task from only 10 state-action pairs.

Table 1: Policy rollout results using ground truth reward for MuJoCo environments using a single demonstration and trained for 1M timesteps. The results are averaged over 10 rollouts and obtained by training the model using five different random seeds.

| Environment | Ant-v3 | HalfCheetah-v3 | Hopper-v3 | Humanoid-v3 | Walker2d-v3 |
|---|---|---|---|---|---|
| Expert | 5159.77 | 8900.85 | 3607.17 | 6249.60 | 4063.81 |
| DAC | $3553.38_{\pm 1580.64}$ | $3292.43_{\pm 1365.91}$ | $3400.87_{\pm 247.88}$ | $169.06_{\pm 40.84}$ | $3433.62_{\pm 362.73}$ |
| GAIL (SAC) | $4261.13_{\pm 415.56}$ | $2569.26_{\pm 1179.47}$ | $2866.05_{\pm 760.34}$ | $147.39_{\pm 59.98}$ | $3296.62_{\pm 417.84}$ |
| GAIL (SAC-SN) | $2956.94_{\pm 697.31}$ | $804.24_{\pm 1441.09}$ | $3263.01_{\pm 540.13}$ | $4643.93_{\pm 2203.33}$ | $3296.62_{\pm 417.84}$ |
| AIRL (SAC) | $361.58_{\pm 1424.61}$ | $-9.47_{\pm 382.93}$ | $3197.09_{\pm 614.25}$ | $4778.57_{\pm 404.82}$ | $3257.18_{\pm 867.42}$ |
| PWIL | $1289.23_{\pm 697.31}$ | $1089.76_{\pm 923.19}$ | $2890.14_{\pm 430.12}$ | $5252.23_{\pm 156.22}$ | $2452.17_{\pm 856.22}$ |
| GAIL (PPO-SN) | $-545.44_{\pm 556.98}$ | $2863.42_{\pm 1155.38}$ | $2859.98_{\pm 1114.94}$ | $425.11_{\pm 125.65}$ | $2648.65_{\pm 1128.32}$ |
| SWIL | $\mathbf{4338.65}_{\pm 555.83}$ | $\mathbf{7481.79}_{\pm 769.22}$ | $\mathbf{3585.28}_{\pm 66.63}$ | $\mathbf{5952.09}_{\pm 315.92}$ | $\mathbf{3936.35}_{\pm 365.69}$ |

## 4.2 ABLATION STUDY

In this section, we perform an ablation study of our approach by considering a number of algorithmic variations of the baselines and our proposed approach. In particular, we address the following settings:

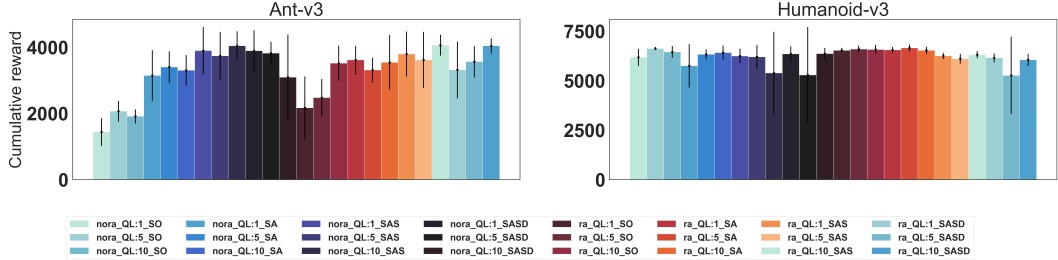

Figure 3: Ablation of SWIL hyperparameters for best evaluation runs on `Ant-v3` and `Humanoid-v3`. The legend format is as follows: *ra* denotes the atom replacement, *ql:x* denotes the number of projections used and *sa* the choice of measurable space. The models are trained using a single trajectory with subsampling factor $\varsigma = 20$.

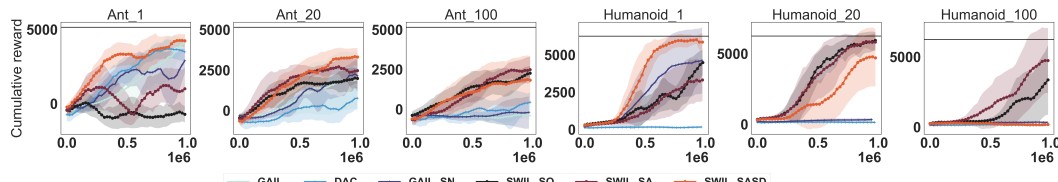

Figure 4: Expert data sparsity ($\varsigma \in \{1, 20, 100\}$) scaling behaviour of algorithms on `Ant-v3` and `Humanoid-v3`. SWIL variants correspond to different $\mathcal{X}$.

- *Underlying measurable space*: we evaluate our method on the following spaces $\mathcal{X} \in \{\mathcal{S}, \mathcal{S} \times \mathcal{A}, \mathcal{S} \times \mathcal{S}, \mathcal{S} \times \mathcal{A} \times \mathcal{S}\}$.

- *Number of projections*: we compare the efficacy of using a moving average of rewards extracted from previous projected $\mathcal{SW}_2$ calculations.

- *Atom replacement*: in this setting, we compare the impact of replacing the projected atoms with samples from the policy.

Figure 3 illustrates the impact of the algorithm variations listed above. We make the following observations. Using measurable spaces with higher dimensionality ($\mathcal{S} \times \mathcal{A} \times \mathcal{S}$ as opposed to $\mathcal{S}$) improves the asymptotic performance significantly for `Ant-v3` but has a modest effect on `Humanoid-v3`. Increasing the number of projections to compute the reward tends to improve average performances and reduce variance for all spaces except $\mathcal{S} \times \mathcal{A} \times \mathcal{S} \times \{0, 1\}$. We ascribe this to the fact that the Cartesian product $\mathcal{S} \times \mathcal{A} \times \mathcal{S}$ takes into account the transition dynamics of the environment, which makes the adversarial optimization more stable. Replacing atoms improves the state-only performance in both cases. Its effect is more pronounced in `Ant-v3`.

## 5 RELATED WORK

**Imitation learning**  Imitation learning via the minimization of $f$-divergences by using a GAN-like approach has been introduced in Ho & Ermon (2016). This method has been extended to the *reward recovery* setting in Fu et al. (2017) by choosing a specific discriminator structure. Ni et al. further extend this to obtain an analytical gradient, while still requiring a discriminator to estimate a density ratio needed to compute the gradient. To combat issues with sample efficiency of on-policy methods used for policy optimization in the above settings, a number of methods have been proposed Kostrikov et al. (2018); Blondé & Kalousis (2019) to alleviate the problem by leveraging policy samples generated by an off-policy agent. The introduction of off-policy samples requires a reinterpretation of the discrepancy measure, as the state occupancy of the policy is no longer defined for a single policy but a mixture of previous iteration samples stored in a replay buffer. Kostrikov et al. reconcile this in a principled way by introducing an optimization scheme which does not require an RL loop. However, the practical instantiation still requires RL samples. Our approach is most similar to off-policy AIL methods like Kostrikov et al. (2018); Blondé & Kalousis (2019) in that it

uses an adversarially learned feature space but does not require regularization and approximates the generalized sliced Wasserstein distance by using a reward based on the $\mathcal{SW}_2$ differential.

**Sliced optimal transport in machine learning** A number of works have leveraged the sliced Wasserstein distance for various machine learning tasks. The authors of Kolouri et al. (2019b) propose to use to regularize an autoencoder architecture by minimizing the distance between prior and posterior measures in the latent space. The $\mathcal{SW}_2$ distance has further been used for implicit generative modeling by defining sliced Wasserstein flows Liutkus et al. (2019). It has further been used for generative modeling using the maximum $\mathcal{SW}_2$ distance in Deshpande et al. (2019). A generalized version of sliced Wasserstein distances has been presented in Kolouri et al. (2019a). We adapt this formulation for the purposes of imitation learning in our work.

**Optimal transport for imitation learning** Optimal transport has previously been explored in the context of imitation and inverse reinforcement learning. The authors of Xiao et al. (2019) make use of the entropy regularized Kantorovich-Rubinstein dual formulation to optimize the policy using a similar scheme to Wasserstein GANs Arjovsky et al. (2017). Dadashi et al. use the primal formulation of the OT problem which relies on a greedy approximation scheme of the true distance. The proposed method relies on the L2 cost in the primal formulation, which is a strong assumption to describe discrepancies in high-dimensional state spaces. Furthermore, the reward is non-Markovian as it relies on the entire dataset of demonstrations. Fickinger et al. (2021) propose an imitation learning method based on the Gromov-Wasserstein distance which allows to compare incompatible spaces across domains. In contrast to these approaches, we propose to use the sliced Wasserstein distance to compute the distance between state occupancy measures using an adversarially learned feature space.

## 6 CONCLUSION

In this work, we have presented a novel imitation learning algorithm which uses the generalized sliced Wasserstein distance to match state occupancy measures between expert and policy samples. The method favorably compares to existing adversarial imitation learning approaches in a number of experimental settings. A number of open research avenues remain. In particular, the current choice of feature space is subject to optimization issues common to adversarial methods. Alleviating this issue would make the method more robust and generalizable. Various regularization strategies might be used such as injecting noise into projection directions using e.g. the variational bottleneck principle **?** could give rise to more diverse policies. Alternatively, autoencoder features or other unsupervised pretraining methods such as contrastive predictive coding Oord et al. (2018) may be investigated. Due to the nonlinear structure of the network, the $\mathcal{GSW}$ distance can only be seen as a pseudo metric owing to the lack of a formal proof for its injectivity. Extending the method to discrete spaces by leveraging e.g. a variant of DQN Mnih et al. (2013) for off-policy optimization would be another valid direction.

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
