# OpenReview forum: "Imitation Learning Using Generalized Sliced Wasserstein Distances"
_ICLR.cc/2024/Conference — ICLR 2024 Conference Withdrawn Submission_

### Official Review · Reviewer_DVGp · 2023-10-29

**Soundness:** 4 excellent
**Presentation:** 4 excellent
**Contribution:** 3 good
**Rating:** 6
**Confidence:** 4

**Summary:**

The paper proposes an imitation learning (IL) method that utilizes the generalized form of the sliced Wasserstein distance. The authors argue that this provides a number of computational and sample complexity benefits compared to existing IL methodologies. The authors match the state occupancy measures between expert and policy samples by using the generalized sliced Wasserstein distance. The authors claim their method showcases state-of-the-art performance compared to other IL algorithms on a number of benchmark tasks from the MuJoCo robotic locomotion suite.

**Strengths:**

- Their problem is well-defined, and methods are explained clearly.
- Related work covers basic prior work.
- The authors properly discuss the methodology used and provide insights about the approaches used therein.
- Their method is compared with many adversarial IL algorithms.

**Weaknesses:**

- While the framework is based upon adversarial IL, it would be interesting to see how the performance of their algorithms compare with the Inverse reinforcement learning algorithms as well as the offline IL algorithms (where there are no further interactions with the environment or the expert). I would be curious to see if there are notable performance improvements over offline IL algorithms because if not, then this limits the practical usage of this algorithms as the interactions could be quite costly.
- The paper does not provide any theoretical guarantees or sample complexity bounds.
- The results are only reported upon MuJoCo tasks. It would be interesting to see the results on other tasks such as MetaWorld manipulation suites, Drone navigation, etc. Given that the paper is mostly empirical, it would be good to have these for completeness.

**Questions:**

Please read the Weaknesses part.

---

### Official Review · Reviewer_jjKQ · 2023-10-30

**Soundness:** 2 fair
**Presentation:** 2 fair
**Contribution:** 2 fair
**Rating:** 5
**Confidence:** 4

**Summary:**

This paper studies imitation learning and inverse RL settings, where the task is to learn a policy that induces a state visitation distribution that is similar to the target distribution defined by trajectories provided by an expert/teacher. This paper proposes to measure, and in turn optimize, the similarity using the Sliced Wasserstein-2 (SW2) distance, due to its appealing computational benefits (from empirical samples) and sample complexity.

**Strengths:**

- The paper provides a good summary of the background literature on optimal transport and Wasserstein distances. That said, as noted in the weaknesses section, this review is too generic and the paper might benefit from a more focused review on how this can be applied more directly to the RL setting, with RL-specific notations.
- The method shows good improvement over baselines, particularly for the "harder" tasks considered in this work (Ant, Humanoid). That said, I have some concerns about the choice of benchmarks studied as outlined in the weaknesses section.

**Weaknesses:**

- **Overly generic notation** While it is commendable that the paper tries to start from the most generic case and ultimately arrives at a practical algorithm by setting various design choices, it might help to have an explicitly written practical algorithm section where the details are clearly spelled out. For example, looking at Algorithm box 1, it is unclear what is the architecture of $g$ and how $\Delta a_\psi$ is computed.
- **Citation format can be improved** Currently the citation style intrudes into the writing, making the reading strenuous. Please consider using \citep{} instead.
- **Impact of the policy optimization algorithm** Can the authors comment on how sensitive the method is to the specific policy optimization algorithm used (SAC in this case). If the claim is that this is a generic IRL algorithm, and primarily about defining a better metric space for rewards in IRL, then evidence of it succeeding with multiple policy optimization methods is necessary in my opinion. Perhaps the authors can consider one on-policy algorithm like PPO and one model-based algorithm like MBPO.
- **Choice of benchmarks** I am a bit concerned about the specific choices of benchmarks, since they are all locomotion tasks that result in repeated or cyclic state visitations. For example, in a 1000-step trajectory with Hopper, there would be several hops within a trajectory that look quite similar (if taken from a high-quality expert policy). As a result, I fear the sample complexity empirical results might be inflated. Can you the authors consider other environments that don't have this property? For example, the Adroit Hand Manipulation Suite [1] could be a good candidate owing to it's high dimensional spaces and the availability of human tele-operated demonstrations, as opposed to trajectories that come from pre-trained expert neural network policies.

[1] Rajeswaran et al. Learning Complex Dexterous Manipulation with Deep Reinforcement Learning and Demonstrations. RSS 2018.

**Questions:**

- Can the method be extended to POMDPs? How about high-dimensional MDPs that are still stateful (e.g. image-based MDPs).
- See comments in the weaknesses section about policy optimization algorithm and choice of benchmarks. Please provide a response to them.

---

### Official Review · Reviewer_BgRr · 2023-10-31

**Soundness:** 3 good
**Presentation:** 3 good
**Contribution:** 2 fair
**Rating:** 3
**Confidence:** 3

**Summary:**

An imitation learning algorithm for minimizing the sliced Wasserstein distance between expert and learned policy. This includes a novel reward function to be optimized by standard RL algorithms for the IL setting.

**Strengths:**

1. Interesting proposition on analyzing the reward and intended goal.
2. Helpful ablations for users of the algorithm to analyze which design choices are important.

**Weaknesses:**

1. The lack of experiments on more challenging tasks (such as dexterous manipulation in Adroit or robomimic, driving in CARLA) in the paper is limiting. Mujoco while standard is also limiting to see the advantages of a new IL algorithm. This is particularly evident from the lack of improvement with SWIL when compared to the best-performing baseline in at least 8-9 of 12 tasks in Figure 2.
2. The sparse setting, while seeming contrived, still only sees SWIL obtain better performance in about 2 or 3 of 5 tasks (the second or third row in Table 1 competes with SWIL when the standard deviation is taken into account).
3. Given the wide array of pre-trained image encoders, it also appears simplistic to only perform experiments with proprioceptive observations. At least 1 or 2 vision-based environments will help prove the scalability of the method. Perhaps CARLA?

**Questions:**

Please see the weaknesses. Additional questions are given below:
1. How could the algorithm can work in a completely offline setting like in offline RL/IL?
2. Can the proposition be proven?
3. Is there an impact of different architectures for the policy? What if a behavior transformer or diffusion policy are used instead?

---

### Official Review · Reviewer_evdi · 2023-10-31

**Soundness:** 3 good
**Presentation:** 2 fair
**Contribution:** 3 good
**Rating:** 5
**Confidence:** 3

**Summary:**

The paper considers the problem of standard imitation learning, and proposes a novel frame work using generalized sliced Wasserstein distances, which presents computational and sample complexity benefits. Based on the approximated differential of the SW2 distance, a per-state reward function is presented to train an RL policy. Experimental results show that the proposed method achieves similar or better performance than the current state-of-the-art IL algorithms on a bench of different tasks.

**Strengths:**

Imitation learning by minimizing generalized sliced Wasserstein distance is a novel and interesting approach. Experimental results indeed show that the proposed algorithm has better sampling complexity than the existing algorithms, and can achieve good performance with very few samples.

**Weaknesses:**

1. The technique writing of the paper has some space of improvement. There are many confusing places in the paper. Please refer to "Questions" for details.

2. The motivation of using generalized sliced Wasserstein distance is not clear to me. In Section 1, it seems that the authors only talk about the SW2 distance is better than the Wasserstein distance in the case of computation, but it is not clear to me why it is better than JS divergence that used in GAIL, or other metrics. In addition, it is not clear why the proposed algorithm has better sample efficiency, it seems the increasing of sample efficiency comes from the off-policy formulation instead of the usage of SW2 distance. I wonder if it can be proved mathematically that SW2 is better than JS divergence or other metric.

3. The presentation of the experimental results are not very clear. For example, in Figure 3, it is hard to compare "nora" with "ra". Please refer to "Questions" for other examples.

**Questions:**

1. In paragraph "MDP", what is the definition of $\Delta(\mathcal S)$ and $\Delta(\mathcal A)$? Why is the input space of $p_\tau(s'\|s,a)$ $\mathcal S\times\mathcal A$ instead of $\mathcal S\times\mathcal A\times \mathcal S$?

2. In paragraph "Imitation learning via distribution matching", what is the definition of $\delta_{s_t^E}$ and $\delta_{s_t^\pi}$? In equation $\rho_e=\sum_{i\leq N}\delta_{s_t^E}$, if the sum is taken on $i$, where is $i$ in $\delta_{s_t^E}$?

3. In paragraph "Optimal transport", where do you define $c(x,x')$ and $\nu(x,x')$?

4. In "Algorithm 1", I guess the loss $\mathcal L$ should be a function of the model parameters instead of the state densities? Also, where is $\phi$ in the loss? What is the difference between $\hat\rho_e$ and $\rho_e$?

5. In Section 6, why there is a question mark after "variational bottleneck principle"?

6. Could the author provide more illustrations on the reason why SW2 is better than other metrics, e.g., JS divergence used in GAIL, in the case of sample complexity and computational efficiency?

7. Could the author provide some intuitions on the reason why AIRL behaves so bad in most of the experiments? In my past experiences, AIRL often performs the similar or better comparing with GAIL.

8. What is the horizontal line in Figure 2 and 4? Is it the reward of the demonstrations?

9. In the experimental results, it seems that increasing number of demonstrations has no effect on SWIL or makes it worse. Could the authors provide some more detailed explanations on this observation?

10. One of the claims of the proposed algorithm is that it has computational benefits. Can the experimental results support this claim?

---

### Official Review · Reviewer_DZ6u · 2023-11-01

**Soundness:** 3 good
**Presentation:** 2 fair
**Contribution:** 2 fair
**Rating:** 3
**Confidence:** 3

**Summary:**

This paper proposes a new imitation learning method based on the sliced Wasserstein (SW) distance. The authors derive a reward function for each state using an approximation to the differential of the SW2 distance, which is compatible with the policy optimization methods in reinforcement learning. Their experiments demonstrate that this new method outperforms existing imitation learning frameworks on various benchmark tasks in the MuJoCo environments.

**Strengths:**

The proposed approach seems to be novel. Empirical results show that the proposed approach is able to perform better than other imitation learning baselines in terms of sample efficiency.

**Weaknesses:**

The writing needs to be improved, especially in section 3.2 which makes the readers hard to follow the reward function design. Figure 1 does not help demonstrate the idea due to the lack of detailed captions.

In the experiment section, the proposed reward design seems to fail when demonstration trajectories are diverse, further analysis is needed.

It seems to me this work is incomplete given the missing proof for Prop. 3.1.

**Questions:**

What's the time complexity of computing the reward function? How does this compare to GAIL and other baselines like PWIL?

What does $x^{+}_t$ and $x^{-}_t$ represents in Figure 1?

Is blue or purple dots expert demonstration samples in Figure 1?

Does the proposed reward resulted in lower variance compare to other baselines during training?

Where is PWIL in Figure 2?

Could the author also include behavior cloning as one of the baselines? (especially for Table 1.)